# *Candida auris* in Austria—What Is New and What Is Different

**DOI:** 10.3390/jof9020129

**Published:** 2023-01-17

**Authors:** Kathrin Spettel, Richard Kriz, Christine Wu, Lukas Achter, Stefan Schmid, Sonia Galazka, Brigitte Selitsch, Iris Camp, Athanasios Makristathis, Heimo Lagler, Birgit Willinger

**Affiliations:** 1Division of Clinical Microbiology, Department of Laboratory Medicine, Medical University of Vienna, 1090 Vienna, Austria; 2Division of Infectious Diseases and Tropical Medicine, Department of Medicine I, Medical University of Vienna, 1090 Vienna, Austria

**Keywords:** *Candida auris*, antifungal resistance, whole genome sequencing, pathogenicity

## Abstract

*Candida auris* is a novel and emerging pathogenic yeast which represents a serious global health threat. Since its first description in Japan 2009, it has been associated with large hospital outbreaks all over the world and is often resistant to more than one antifungal drug class. To date, five *C. auris* isolates have been detected in Austria. Morphological characterization and antifungal susceptibility profiles against echinocandins, azoles, polyenes and pyrimidines, as well as the new antifungals ibrexafungerp and manogepix, were determined. In order to assess pathogenicity of these isolates, an infection model in *Galleria mellonella* was performed and whole genome sequencing (WGS) analysis was conducted to determine the phylogeographic origin. We could characterize four isolates as South Asian clade I and one isolate as African clade III. All of them had elevated minimal inhibitory concentrations to at least two different antifungal classes. The new antifungal manogepix showed high in vitro efficacy against all five *C. auris* isolates. One isolate, belonging to the African clade III, showed an aggregating phenotype, while the other isolates belonging to South Asian clade I were non-aggregating. In the *Galleria mellonella* infection model, the isolate belonging to African clade III exhibited the lowest in vivo pathogenicity. As the occurrence of *C. auris* increases globally, it is important to raise awareness to prevent transmission and hospital outbreaks.

## 1. Introduction

Infections with *Candida* species are a severe public health problem across the world, especially since the emergence of *Candida auris,* a multidrug-resistant (MDR) nosocomial pathogen that can cause invasive candidiasis and is often associated with a high case fatality rate and therapeutic failure [1]. *Candida auris* was first isolated and described from ear discharge of a 70-year-old female patient in Tokyo, Japan, in 2009 [2]. Within a decade, this yeast had spread around the world at an alarming rate, causing numerous outbreaks in health care facilities [1,2,3,4,5,6,7,8,9].

*C. auris* has an extraordinary capacity to colonize the human body and environment for prolonged periods [10] and is very difficult to eradicate [11]. During the current COVID-19 pandemic, many health care systems around the world were overwhelmed, leading to over-occupied intensive care units (ICUs) and compromised infection prevention control. These circumstances might favor the nosocomial spread of *C. auris* infections [12,13]. Furthermore, *C. auris* is often misidentified by commercial identification systems in routine microbiology laboratories; therefore, the real prevalence of infections caused by *C. auris* may be underestimated.

Here, we describe phenotypic and genotypic features of the first five Austrian isolates. Given the rapid global emergence of *C. auris* and the associated hospital outbreaks, it is important to raise awareness about the appearance and possibility of outbreaks of *C. auris* in Austrian healthcare facilities.

The aim of this study was to describe the morphological characteristics and to investigate the isolates’ susceptibilities to commonly used antifungals, including the new antifungals ibrexafungerp and manogepix. In addition, mutations associated with antifungal resistance were investigated. A *Galleria mellonella* infection model was chosen to determine the in vivo pathogenicity of the five isolates. Finally, the study aims to assign the five strains to one of the four previously described phylogenetic clades of *C. auris*.

## 2. Materials and Methods

### 2.1. Sampling

Five isolates were detected in patients from Austria and sampled. In addition to the five clinical isolates, we included two control strains, CBS 12777 (South Asian clade I) and CBS 10913 (alias: B11220, first East Asian clade II from Japan [2]), in the phenotypic characterization (susceptibility testing, aggregating phenotype, in vivo pathogenicity). However, as there are no reference values for most of the data, these were included as type strains for comparison with the data obtained from the clinical isolates.

### 2.2. Phylogeographic Clade Assignment Using Whole Genome Sequencing (WGS)

DNA extraction was carried out using 48 h colonies from malt extract agar (MEA) plates, following a solution-based protocol with implemented bead-beating steps as described elsewhere [14]. Before preparing libraries, the total quantity of DNA was measured using the Qubit dsDNA High Sensitivity Assay system (Life Technologies, Carlsbad, CA, USA). The purity of DNA was determined by measuring the ratios A_260_/A_280_ and A_260_/A_230_ using a NanoDrop 2000c spectrophotometer (Thermo Fisher Scientific, Waltham, MA, USA).

In this study, WGS was carried out to classify the clinical isolates according to the four major phylogeographic clades. DNA-sequencing libraries were prepared from extracted DNA according to the Illumina DNA Prep workflow. DNA was denatured according to the protocol and diluted to a final loading concentration of 8 pM combined with a 5% PhiX spike-in (PhiX Control v3, Illumina) for sequencing on a v3-flowcell 2 × 300 bp on Illumina MiSeq (Illumina, San Diego, CA, USA).

The quality of the NGS run was evaluated with FastQC 0.11.4 [15]. The Mycosnp pipeline was used for phylogenetic analysis of the WGS data [16]. The sequence of the *C. auris* isolates were compared to B8441 (South Asian clade I) and B11221 (African clade III), which were obtained from www.candidagenome.org (accessed on 17 September 2022), and B11220 (East Asian clade II, GCA_003013715.2) and B11243 (South American clade IV, GCA_003014415.1), obtained from www.ncbi.nlm.nih.gov (accessed on 17 September 2022). For phylogenetic analysis, we compared SNPs called against the reference strains B8841, B11220, B11221 and B11243. A multiple sequence alignment of mating type loci and flanking genes of *C. auris* isolates Cau1-5 with B8441 (*MTLa*) and B11221 (*MTLα*) was generated to determine mating type [17].

### 2.3. Microscopic Morphology—Aggregating or Non-Aggregating Phenotype

Cellular appearance of all isolates was examined after 48 h of incubation at 37 °C in a liquid medium to determine whether aggregate formation is inducible in any of the *C. auris* strains through exposure to antifungal drugs. All isolates were microscopically screened for aggregating phenotype in a RPMI medium, Sabouraud medium and after exposure to different antifungals (caspofungin 0.016 mg/L, fluconazole 0.25 mg/L, 5-flucytosine 0.064 mg/L and amphotericin B 0.064 mg/L). All microscopic examinations, including aggregate-forming capacity after antifungal exposure, were performed in triplicates for all five clinical isolates, plus control strains CBS 10913 and CBS 12777, on three different days.

### 2.4. Infection Model in Galleria mellonella

For the infection model, *G. mellonella* larvae were used in a weight range of 250 mg ± 30 mg and after a day of fasting. Before injecting with the different *C. auris* strains, the larvae were washed with 96% ethanol to minimize the surface pathogens. The larvae were split into groups of 10 and kept in petri dishes. *C. auris* strains were cultured overnight on Sabouraud agar and resuspended in sterile PBS. Using the cell counter Luna-FL (Logos Biosystem), a cell suspension with a concentration of 5.0 × 10^8^ CFU/mL was obtained for each strain. For the trial, each group of larvae was inoculated with 10 µL of the *C. auris* suspension via injection in the last left pro-leg using a sterile Hamilton syringe (Hamilton Medical, Swiss). Thus, each larva was inoculated with 5.0 × 10^6^ yeast cells. A control group of 10 larvae was injected with 10 µL of PBS. Afterwards, the larvae were incubated at 37 °C and at different time points (12, 24, 48, 72, 96 and 120 h), the melanization and movement of the larvae were visually checked to evaluate survival. For comparison of pathogenicity, the median effective dose(ED50) values, which represent the time in hours after which 50% of the inoculated larvae population died, were calculated.

### 2.5. Antifungal Susceptibility Testing (AFST) including New Antifungals Ibrexafungerp and Manogepix

Antifungal susceptibility of all clinical isolates and the two control strains was determined according to the European Committee on Antimicrobial Susceptibility Testing (EUCAST) E.Def 7.3.2 microdilution method [18]. Each isolate was tested against the following eight antifungals (values represent the final concentrations after inoculation): anidulafungin (ANI; 0.008–16 mg/L), micafungin (MCA; 0.008–16 mg/L), fluconazole (FLC; 0.125–256 mg/L), posaconazole (POS; 0.016–32 mg/L), voriconazole (VOR; 0.008–16 mg/L), 5-flucytosine (5-FC; 0.032–64 mg/L), amphotericin B (AMB; 0.032–16 mg/L), ibrexafungerp (IBX; 0.016–8 mg/L) provided by Scynexis (Jersey City, NJ, USA), and manogepix (MGP; 0.002–16 mg/L) provided by Pfizer (New York City, NY, USA). There are no established species-specific clinical breakpoints for *C. auris* in EUCAST guidelines [19].

## 3. Results

### 3.1. Origin of Candida Auris Isolates

To date, five sporadic cases of *C. auris* have been reported in Austria—one infection and four cases of asymptomatic colonization. The first *C. auris* strain (Cau1) was isolated in January 2018 from the external auditory canal of a 22-year-old man who suffered from a therapy-refractory otitis externa that had persisted for almost four years despite antibiotic treatment. The patient was otherwise healthy. He was of Turkish ancestry and used to travel to Turkey frequently [20]. The second isolate (Cau2) was isolated in February 2020 from the external auditory canal of a 61-year-old male with a hematologic disease, followed by a strain (Cau3) isolated in May 2020 from the urinary tract of a trauma patient of Indian origin with regular travel activity. The isolate Cau4 was detected in October 2021 from a 60-year-old female patient in Valencia, Spain, after hospitalization because of a subarachnoid hemorrhage. The most recent isolate (Cau5) was detected in April 2022 in the urine of a 66-year-old female after hospitalization on Patmos and Rhodos, Greece, also due to subarachnoid hemorrhage. The four latter patients were colonized and did not show any signs of infection caused by *C. auris*. All patients came from different regions in Austria and were treated in different healthcare institutions. No epidemiologic links could be found between these five cases. In Table 1, all five clinical *C. auris* isolates are listed, including patient description, underlying medical condition, isolation site and origin of the isolate.

### 3.2. Phylogeographic Clade Assignment

Phylogenetic analysis using WGS revealed that the isolates Cau1, Cau2, Cau3 and Cau5 were genetically closest to South Asian clade I (see Table 2). Isolates Cau1 and Cau2, as well as Cau3 and Cau5, showed a higher degree of relatedness to each other with 44 SNPs and 76 SNPs, respectively. Overall, it is likely that the strains Cau1 and Cau2 belong to different subclades compared to Cau3 and Cau5, and reference strain B8441, as they are separated by more than 200 and 900 SNPs, respectively. In addition, the isolates could be separated from clades II, III and IV by 40,000 to 170,000 SNPs, whereas the isolate Cau4 was most closely related to African clade III, with 49 SNPs to clade III reference isolate B11221. Furthermore, analysis of the mating-type locus revealed that the clinical isolates Cau1, Cau2, Cau3 and Cau5 were *MTLa* homozygous (*MTL*a1 and *MTL*a2), thus confirming their assignment to South Asian clade I. The isolate Cau4 showed a homozygous *MTLα*, as is expected from African clade III.

### 3.3. Aggregating Phenotype of C. auris

When examining the cellular morphology of *C. auris* grown for 48 h at 37 °C in RPMI media, the isolates Cau1, Cau2, Cau3 and Cau5, as well as the two control strains CBS 12777 and CBS 10913, appeared as individual budding yeast cells. Isolate Cau4—assigned to African clade III—was forming aggregates in RPMI media and after exposure to different antifungal agents. Isolate Cau1 showed a non-aggregating phenotype in RPMI media, but formed small to large aggregates reproducibly, which could not be physically disrupted when exposed to different antifungals—see Figure 1.

### 3.4. Infection Model with Galleria mellonella

The *G. mellonella* infection model showed that for all tested *C. auris* isolates, a concentration of 5.0 × 10^8^ CFU/mL (5.0 × 10^6^ CFU/larva) is sufficient to kill *G. mellonella* within 120 h.

All of the PBS-injected larvae which were used as a negative control survived the experiment. Figure 2 and Table 3 demonstrate the different pathogenicity of the clinical isolates in comparison to the control strains. As shown in Table 3, the strains CBS 10913 and Cau4 are less pathogenic than the others.

### 3.5. Antifungal Susceptibility

In vitro susceptibilities of the five clinical *C. auris* isolates and two control strains to nine different commonly used antifungal agents and the new antifungals ibrexafungerp and manogepix are displayed in Table 4. All five isolates exhibited high MIC values to at least two different classes of antifungals. Both isolates Cau1 and Cau2 showed moderately elevated MIC values to the tested echinocandins anidulafungin (0.5 mg/L) and micafungin (0.125 mg/L)—as well as to amphotericin B (2 mg/L). Fluconazole showed low in vitro efficacy against isolates Cau3 and Cau5 with MIC values of 64 and >256 mg/L, respectively. The isolate Cau4 displayed a high fluconazole MIC of 64 mg/L as well as slightly elevated MIC values against echinocandins. The new antifungal substances ibrexafungerp and manogepix showed good in vitro activity against all tested isolates. The MIC values of manogepix were particularly low, with 0.008–0.032 mg/L.

### 3.6. Detection of Antifungal Resistance Mutations

No mutations in the genes *ERG1*, *ERG2*, *ERG3* (associated with azole resistance), and *MEC3* (associated with polyene resistance) could be detected in any of the clinical isolates [21]. In the isolates Cau3 and Cau5 showing high fluconazole MICs, the missense mutation p.Y132F in *ERG11*, which encodes 14-alpha-demethylase, was detected. The isolate Cau4 (African clade III) had two missense mutations (p.V125A, p.F126L) in *ERG11* compared to the reference sequence of B8441 (South Asian clade I) and B11220 (East Asian clade II). Additionally, the isolates Cau3 and Cau5 had two missense mutations p.A583S (Cau3, Cau5) and p.S857L (Cau5) in *TAC1b*, which is a transcription factor for the drug efflux pumps Cdr1p, Cdr2p and Snq2p. Furthermore, the missense mutation p.F1367C was found in *FKS1* in Cau1 and p.K74E (Cau1, Cau2, Cau3, Cau5) in *CIS2*. No mutations compared to the used reference genome of B11221 (African clade III) were found in the isolate Cau4.

## 4. Discussion

*C. auris* is a very heterogeneous, drug-resistant yeast that has been recognized as a serious health threat. Though *C. auris* has attracted much scientific attention, there are still many open questions. Therefore, the isolates detected in Austria have been investigated thoroughly in order to obtain more knowledge regarding the characteristics of this fungus and to check the strains for differences compared to previously described isolates. Thus, different methods, such as whole genome sequencing for classification of the phylogenetic clade, conventional phenotyping, assessing the pathogenicity an in vivo model, and the susceptibility pattern, as well as detecting resistance mutations, were applied and will be discussed in this order.

### 4.1. Phylogeographic Clades

In the present study, WGS of the five clinical isolates detected in Austria was conducted to determine their phylogeographic origin. The isolates Cau1, Cau2, Cau3 and Cau5 were separated by 46 to 225 SNPs, confirming the assumption that there are no epidemiological links between them. They were genetically closest to South Asian clade I and could be separated from clade II, III and IV by 45,000 to 165,000 SNPs. The patient colonized with Cau4 belonging to the African clade III was hospitalized in Valencia due to a subarachnoid hemorrhage. Since a large nosocomial outbreak with *C. auris* belonging to the African clade III has been described in a hospital in Valencia, it can be speculated that the patient acquired colonization during her hospital stay [22]. The patient colonized with Cau5 belonging to South Asian clade I was hospitalized on Patmos and Rhodos in Greece due to subarachnoid hemorrhage. From 2019–2021 74 cases and outbreaks in Greece have been reported including 58 cases in 2021, thus indicating a significant increase of cases in Greece. However, no further information which can be linked to our case has been made available [23].

Analysis of the mating-type locus—for further validation of the phylogenetic classification—revealed that our South Asian clade I isolates were *MTL*a homozygous (*MTL*a1 and *MTL*a2), supporting their assignment to South Asian clade I. In contrast to the other isolates, isolate Cau4 showed a homozygous *MTLα* as is described for African clade III strains [17].

The South Asian clade I is known for high rates of resistance to different antifungal agents and has been linked to invasive candidiasis and large-scale hospital outbreaks. However, isolates Cau1 and Cau2, belonging to South Asian clade I, may not explicitly display all of the properties associated with this clade. Both isolates were detected from the external auditory canal of two different patients—one with infection and one being only colonized. *C. auris* isolates associated with ear infections usually belong to East Asian Clade II. Welsh et al. indicate that this observation may be biased due to different testing methods. For example, in many laboratories, species-level identification is rarely performed from non-sterile sites such as the ear canal. Nonetheless, East Asian clade II is the only clade not associated with hospital outbreaks of invasive candidiasis [24].

The four *C. auris* clades can differ in their susceptibility profiles significantly. East Asian clade II is usually considered more susceptible than South Asian clade I. Although the isolates Cau1, Cau2, Cau3 and Cau5 belong to South Asian clade I, only Cau3 and Cau5 are fluconazole-resistant, while in previous studies, South Asian clade I has been shown to have a very high resistance rate of 97% to fluconazole [25]. However, levels of antifungal drug resistance can vary among strains significantly [24,26]. Nevertheless, four of our isolates showed elevated MIC values against amphotericin B (2–8 mg/L), a characteristic which, to date, has been found only in clades I and IV. Furthermore, all five isolates showed slightly elevated MICs for echinocandins, which is also in agreement with previous studies that reported cases of echinocandin resistance in clades I, III and IV [25,27].

### 4.2. Aggregating Phenotype

To date, only a few studies have investigated aggregate formation in *C. auris.* Borman et al. found that *C. auris* isolates can be roughly divided into aggregative or non-aggregative groups based on their growth characteristic phenotypes in culture. Isolates with an aggregative phenotype form large cellular agglomerates which cannot be physically disrupted. This is caused by a malfunctioning budding process, after which the daughter cells are not released. Aggregate-forming isolates were shown to be less pathogenic than isolates with non-aggregative phenotypes in a *G. mellonella* infection model [28]. Szekely et al. observed that differences in phenotypic behavior of *C. auris* isolates were clade-specific—including their growth characteristics and antifungal susceptibility profiles. In this study, aggregate-forming isolates belonged exclusively to the African clade III (n = 50) and not a single isolate from the South Asian clade I (n = 90) showed this phenotype. The authors also noted that an exposure to echinocandins and azoles caused isolates from the South Asian clade I to grow as aggregates. Cell morphology was unaffected by exposure to flucytosine or amphotericin B [29]. In isolate Cau4, belonging to the African clade III, the aggregating phenotype was detected. The other four clinical *C. auris* isolates belonging to South Asian clade I were non-aggregative: they appeared as single cells when cultured in standard mycological media. When the clinical isolates were exposed to antifungal drugs, the isolate Cau1 switched to the aggregative phenotype. However, in contrast to the findings of the previously mentioned study by Szekely et al., aggregation was also observed after exposure to flucytosine or amphotericin B. Aggregation might be an escape, stress response or a physical defense against antiseptic or antifungals in naturally non-aggregative *C. auris* isolates [29]. However, the reason for these observed differences in phenotypic behavior remains unclear, since the antifungal susceptibility profile of Cau1 did not differ from the other three South Asian clade I isolates in which aggregates were not detected. Further investigations are required to evaluate whether such in vitro differences in phenotypic behavior have implications for clinical practice.

### 4.3. Pathogenicity in the Galleria mellonella Infection Model

For the evaluation of pathogenicity, a *G. mellonella* infection model was performed. It became apparent that the four clinical isolates belonging to the South Asian clade I and the control strain CBS 12777 showed a much lower ED50. The isolate Cau4, which could be assigned to the African clade III, as well as the control strain CBS 10913 (East Asian clade II), showed a higher ED50 of 57.31 and 44.46, respectively. As already mentioned, aggregation was observed in isolate Cau4, which is in accordance with previous reports suggesting that aggregating isolates are less pathogenic in in vivo models in contrast to South Asian clade I isolates [28,30].

### 4.4. Antifungal Susceptibility

As EUCAST has not established species-specific clinical breakpoints for *C. auris* yet, an exact interpretation is difficult. All five clinical isolates of *C. auris* exhibited high MIC values to at least two different classes of antifungals. Since only four drug classes are available for the treatment of invasive *Candida* infections, this could limit therapeutic options.

In accordance with previous studies that reported the efficacy of the pyrimidine analogue 5-FC against *C. auris* [31,32,33], all five isolates showed low 5-FC MIC values of 0.064–0.25 mg/L. However, 5-FC is not used as monotherapy because in vivo resistance develops rapidly during treatment [34]. *C. auris* shows an extraordinarily high rate of fluconazole-resistant strains compared to other *Candida* species [1,35,36]. Therefore, echinocandins are usually the drug of choice for treatment of *C. auris* infections [37]. Nevertheless, the results of this study demonstrate that echinocandin resistance can develop in *C. auris*, thus confirming previous reports [4,26,38]. Cau1 and Cau2 were susceptible to azoles and showed moderately increased MIC values to both echinocandins and amphotericin B. In Cau3, the MIC of fluconazole was 64 mg/L, in addition to moderately elevated MIC values against echinocandins and amphotericin B. Moreover, the isolate Cau4 showed a high fluconazole MIC (64 mg/L). In contrast to the other strains, Cau4 was susceptible to anidulafungin and amphotericin B. The most recent isolate, Cau5, showed a similar resistance profile to that of Cau3, although fluconazole and amphotericin B MICs showed higher MIC values. These findings demonstrate the difficulty in choosing the appropriate antifungal treatment and the need for antifungal susceptibility testing. Furthermore, this highlights the urgent need for new antifungal drugs that may be effective against *C. auris*. The MIC values regarding the new antifungals ibrexafungerp and manogepix showed excellent in vitro activity, suggesting a promising alternative for an effective therapy, as has already been described [39,40,41,42,43]. In particular, manogepix showed potent in vitro activity with MIC values of 0.008–0.032 mg/L. This supports the assumption that manogepix could be a valuable therapeutic alternative in the treatment of *C. auris* [44,45,46].

### 4.5. Screening for Potential Resistance Mutations

Molecular mechanisms of antifungal resistance are not completely understood—this is especially true in the case of *C. auris*. However, NGS has been proven to be a powerful tool to detect mutations that are associated with antifungal resistance in *Candida* spp. [47,48]. When examining the five clinical isolates, a mutation in *ERG11* (p.Y132F; hot spot 1 region), one of the most popular mutations associated with azole resistance in *C. auris* [49], was detected in Cau3 and Cau5. This was correlated with the elevated MIC values of 64 mg/L and >256 mg/L for fluconazole, respectively. The isolate Cau4 belonging to the African clade III, with a MIC of 64 mg/L for fluconazole, showed two missense mutations in *ERG11* (p.V125A, p.F126L) compared to the reference genomes of clades I and II. Since this location is only six amino acids upstream of the well-known mutation p.Y132F, these target mutations may be associated with azole resistance. However, the absence of any mutations in *ERG11* does not rule out the presence of azole resistance in *C. auris*. Kwon et al. reported that only in 5 of 38 fluconazole-resistant East Asian clade II *C. auris* isolates amino acid substitutions in *ERG11* could be detected [50].

Regarding echinocandin resistance, Cau1 showed a mutation (p.F1367C) ten amino acids downstream of the *FKS1* gene hot spot 2, which could explain the moderate resistance to echinocandins. However, Cau2, Cau3, Cau4 and Cau5 did not show any mutations in *FKS1* despite elevated echinocandin MICs, thus suggesting other underlying molecular resistance mechanisms. WGS analysis revealed a missense mutation in *CIS2* (p.K74E), which could be found in all South Asian clade I isolates (Cau1, Cau2, Cau3, Cau5). The *CIS2* gene encodes gamma-glutamyl transpeptidase (GGT) and has been recently suspected to play a role in echinocandin resistance in *C. auris* [21]. Carolus et al. found that decreased amphotericin B susceptibility accompanied by fluconazole cross-resistance was caused by simultaneous premature stop codons in *ERG3* and *ERG11* [21]. Previously, amphotericin B resistance was only explained by an increased expression of genes involved in the biosynthesis of ergosterol, such as *ERG1* and *ERG2* [17]. The Austrian isolates did not show any mutations in *ERG1, ERG2* and *ERG3.*

*C. auris* is still understudied compared to *C. albicans* and *C. glabrata,* even though it poses an urgent global threat and has been recently included in the Critical Priority Group of the WHO fungal priority pathogens list [51]. Further research is needed in order to understand the diverse underlying molecular resistance mechanisms associated with antifungal resistance, as this might aid in the development of new drugs [21,37].

## 5. Conclusions

More than thirteen years after the first description of *C. auris* in Japan [2], there are still numerous open questions. For example, little is known about its mechanisms and the development of resistance to antifungals.

The present study underlines the importance of raising awareness among healthcare providers of a potential spread of *C. auris* in Austria as the incidence of *C. auris* infections increases worldwide. All five clinical isolates being detected in Austrian patients show in vitro multidrug resistances. Four isolates belong to the South Asian clade I and one isolate could be assigned to African clade III. Both genotypical lineages have been associated with invasive infections and high case fatality rates [1]. It is conceivable that there are more cases of *C. auris* in Austria that have neither been detected nor reported. It is important to ensure a rapid identification of *C. auris* in patients and develop awareness of the threat this yeast may cause in order to prevent possible transmissions and hospital outbreaks.

## Figures and Tables

**Figure 1 jof-09-00129-f001:**
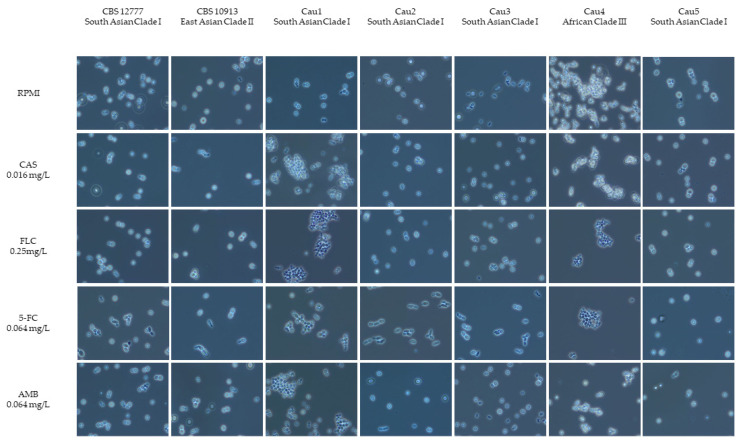
Cellular morphology of *C. auris* isolates. Microscopic appearance of *C. auris* cells grown in RPMI and by exposure to antifungal agents (CAS—caspofungin 0.016 mg/L), FLC—fluconazole (0.25 mg/L), 5-FC—5-flucytosine (0.064 mg/L), AMB—amphotericin B (0.064 mg/L). Suspensions were examined at 400× magnification by phase contrast microscopy.

**Figure 2 jof-09-00129-f002:**
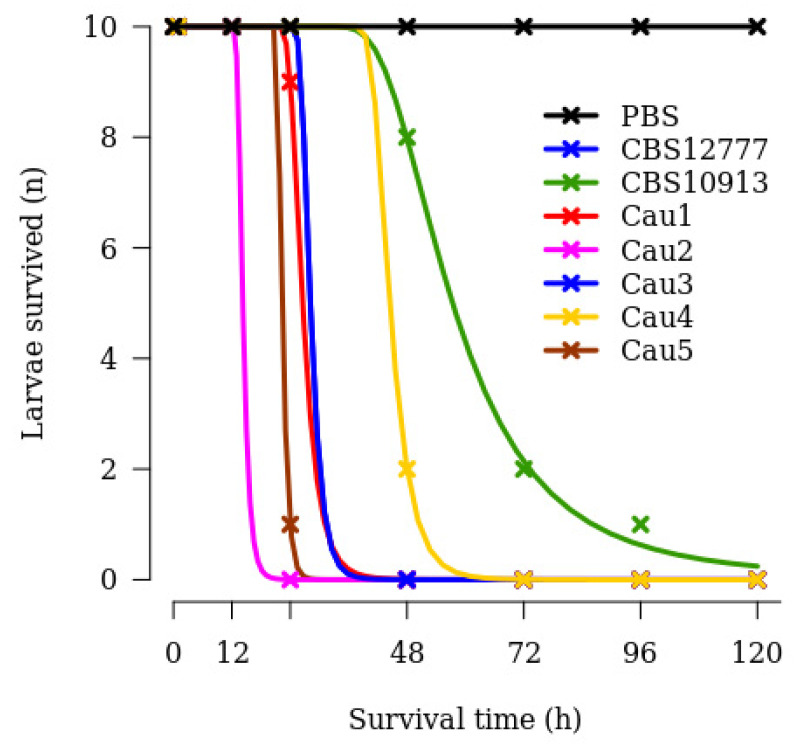
Time–kill curves of the different *C. auris* strains in the *G. mellonella* infection model with a concentration of 5.0 × 10^6^ CFU/larva and PBS as a control group.

**Table 1 jof-09-00129-t001:** Description of the clinical isolates detected in Austria.

ID	Isolation Date	Patient Description	Underlying Medical Condition	Site of Isolation	Travel History
**Cau1**	01/2018	22-year-old male patient with Turkish ancestry	therapy-refractory otitis externa	external auditory canal	Turkey
**Cau2**	02/2020	61-year-old male patient	hematologic malignancy, colonization	external auditory canal	none
**Cau3**	05/2020	male patient with Indian ancestry	trauma, colonization	urinary tract	India
**Cau4**	10/2021	60-year-old female patient	hospitalization in Spain due to subarachnoid hemorrhage, colonization	throat	Spain
**Cau5**	04/2022	66-year-old female patient	hospitalization in Greece due to subarachnoid hemorrhage, colonization	urinary tract	Greece

**Table 2 jof-09-00129-t002:** Number of SNPs detected in the Austrian *C. auris* isolates.

Isolate	Cau1	Cau2	Cau3	Cau4	Cau5	B8441 (clade I) South Asia	B11220 (clade II) East Asia	B11221 (clade III) Africa	B11243 (clade IV) South America
**Cau1**	0	44	220	42240	225	959	63,691	44,333	163,671
**Cau2**	44	0	204	42294	209	959	63,736	44,419	163,671
**Cau3**	220	204	0	42314	76	978	63,658	44,359	163,983
**Cau4**	42,240	42,294	42,314	0	42,300	42332	59,809	49	157,871
**Cau5**	225	209	76	42300	0	932	60,184	42,259	155,590

**Table 3 jof-09-00129-t003:** ED50 values of different *C. auris* strains in the *G. mellonella* infection model, representing the time in hours after which 50% of the larvae population died.

Sample ID	ED50	SE	CI (Lower)	CI (Upper)	Events
**PBS control**	NA	NA	NA	NA	0
**CBS12777**	28.15	0.19	27.62	28.67	10
**CBS10913**	57.31	1.28	53.76	60.86	10
**Cau1**	26.56	0.11	26.25	26.87	10
**Cau2**	14.25	0.12	13.9	14.6	10
**Cau3**	28.15	0.19	27.62	28.67	10
**Cau4**	44.46	0.18	43.96	44.97	10
**Cau5**	22.35	0.05	22.22	22.47	10

SE = standard error, CI = confidence interval, Events = total number of died larvae, NA = not applicable.

**Table 4 jof-09-00129-t004:** In vitro susceptibility pattern of *C. auris* isolates tested by EUCAST microdilution method. MIC—minimum inhibitory concentration.

Antifungal Agent		Clinical Isolates					Control Strains	
		Cau1	Cau2	Cau3	Cau4	Cau5	CBS 10913	CBS 12777
		MIC (mg/L)						
** *Echinocandins* **	*Anidulafungin*	0.5	0.5	0.064	0.032	0.25	0.032	2
	*Micafungin*	0.125	0.125	0.125	0.064	0.25	0.064	0.5
** *Triazoles* **	*Fluconazole*	0.5	2	64	64	>256	8	>256
	*Posaconazole*	0.032	0.032	0.032	≤0.016	0.125	0.032	0.25
	*Voriconazole*	0.008	0.032	0.125	0.5	1	0.064	1
** *Pyrimidine analogues* **	*5-Flucytosine*	0.064	0.125	0.125	0.25	0.125	0.125	0.125
** *Polyenes* **	*Amphotericin B*	2	2	4	1	8	1	2
** *New Antifungals* **	*Ibrexafungerp*	0.064	0.125	0.125	0.125	0.25	0.032	0.25
	*Manogepix*	0.008	0.008	0.008	0.008	0.032	0.008	0.032

## Data Availability

The sequencing data are available in the BioProject database under accession number PRJNA923734.

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
