# Peer review of "Candida auris in Austria—What Is New and What Is Different"

_jof, 2023, doi:10.3390/jof9020129_

Round 1

Reviewer 1 Report

Major comments

1) The authors used WGS to identify variations between C. auris isolates. In 2021 a “Whole-Genome Sequence Benchmark Dataset for Phylogenomic Pipelines for Candida auris” was published (DOI: 10.3390/jof7030214), which allows validation of a developed WGS pipeline. In order to compare the obtained WGS results with other WGS data on C. auris, the authors should validate their WGS results with this benchmark dataset and also show their WGS differences as SNPs and not as variants, which also includes indels. Clade I isolates mostly differ less than 80 SNPs (see also comment 4).

2) In Table 2 and 3 and Figure 1 the authors show the effect of injecting different C. auris strains in G. mellonella. For improved clarity Table 2 and 3 should be combined in one Table and Figure 2, in which the fonts are much too small, should be used as Supplementary Figure.   

3) Antifungal susceptibility testing: the authors mention the use of two CBS strains as control strains in the AFST test (lines 195-196). During the rest of this paragraph the “control” strains are discussed as they are part of the experiment (“all seven isolates”), and not controls. The authors also do not mention whether AFST data of the controls are in line with expected data. As such they do not function as controls. Please adapt.  

4) Lines 248-249: The authors point out that the clade I variants show “very similar numbers” and “are highly related”, while they differ by at least 605 variants. Different articles demonstrate that the number of SNPs between the majority of clade I variants is less than 80 SNPs (DOI: 10.1128/spectrum.02645-22 and DOI: 10.3390/jof7030214). Adapt after validating WGS results and showing results as SNPs, as indicated under comment “1”.   

5) The discussion section should be shortened.

Minor comments:

Line 136: “Bowtie2 2.2.7.  was used to assemble”. This program is only used for mapping.  

Line 168-170: These lines refer to a concentration that is sufficient to kill, however the subject it does not mentioned. I guess the authors mean CFU of C. auris? It is also not clear whether this is an observation in this study (then also mention which isolate) or a previous study (then put reference).  

Line 232: “has” should be “had”

The identification of MTLa1 and a2 is not described in M&M.

Lines 319-330: the authors often use phrases like “are able to”, “could also be induced”, “could (not) be induced”. These should be avoided.

Lines 361: delete “and possibly resistant”

Line 362: delete “definitive”. 

Reviewer 2 Report

Summary:

This manuscript provides a thorough characterization of the five clinical C. auris isolates recovered to date in Austria.  The authors did an excellent job in providing relevant details pertaining to origin (geographic, body site, etc.), susceptibility (MICs), genotype (WGS), phenotype (aggregation), and virulence (G. mellonella infection model).  The manuscript is well written and would be informative in the documentation of the continued global spread of this problematic Candida pathogen.  An opportunity for improvement comes primarily in changing the approach of using C. albicans susceptibility trends and ECOFFs/BPs as context for interpretation of the C. auris data generated.

Major comment:

Susceptibility testing data/Table 4 content: While it is informative to have some sort of context in which to view the susceptibility data, the use of C. albicans is not an ideal Candida spp. in this capacity.  Not only are C. albicans MIC values for echinocandins (eg most relevant first line agents for C. auris) much lower than for C. auris, but C. albicans does not have a similarly high predisposition to antifungal drug resistance as C. auris either.  Some thoughts:

-        - C. glabrata could be used in place of C. albicans (eg applying EUCAST ECOFFS and BPs) -- at least it has somewhat higher candin MIC values (although still multiple dilutions lower than C. auris), elevated incidence of drug resistance, and a shared haploid genome.

-        - Alternatively, no comparative Candida spp. ECOFF/BP data could be shown in Table 4 and the study data could be spoken to in Results/Discussion in the context of published C. auris susceptibility data for both new agents.

-       -Typically, in the absence of BPs, ECOFFS are used, while those are not available here either, there are a number of C. auris EUCAST studies on approved antifungals (as well as ibrexafungerp and manogepix) that provide valuable data for reference to help inform whether the 5 isolates are WT/NWT.  Two C. auris EUCAST MIC studies using a 122-strain panel have been completed which provide WT-UL/ECOFF analyses that would be very informative to the reader and help strengthen this manuscript.  Both of these are Arendrup et al 2020 AAC and the manogepix pub is cited already (# 46) but the ibrexafungerp one is not but should be (PMID 31844005; I have no affiliation or COI with either of these publications). 

-Abstract, Results, Discussion: I would suggest based on the above comments to reword any comparative activity statements and limit inferences of resistance to the context of established C. auris data for each agent/class.

-Caspofungin: EUCAST does not support susceptibility testing of caspofungin – I’m not sure if presentation of these MIC data is problematic in that sense.   

Minor comments:

Lines 83-86: How/why were the concentrations selected for each agent?  Given that each strain has a different susceptibility profile, is there any limitation by keeping these incubation concentrations the same for each vs. as a defined fraction of the MIC?

Lines 116-117: The sources for ibrexafungerp and manogepix should be stated.

Lines 147-149: The collection from which the CBS control strains were obtained should be cited.

General: Check for consistency throughout the ms for the CBS strains as to whether there is a space between “CBS” and the strain # or not and modify to whichever the correct format is.

Reviewer 3 Report

The present study describes five C. auris isolates detected in Austria. For four of the five isolates possible abroad origin was suggested,  despite It is very probable that there are more cases of undetected C. auris in Austria this may indicate that the incidence of Candia auris in Austria could still be low and this study could contribute to creating awareness among health care providers about the possible spread and also encourage epidemiological surveillance programs to detect cases of C. auris colonization, mainly in the hospitals where these isolates were collected.

It is remarkable that the study addresses the description of the isolates from various approaches, infection model, WGS, and antifungal susceptibility profiles thus making the significance of the findings of the manuscript appropriate for publication in the Journal of Fungi

The table heatmap of the graphical abstract figure is missing a legend describing the colors and a note mentioning that C. albicans breakpoints were used

Line 17, The authors may want to consider the tentative MIC values proposed by the CDC (https://www.cdc.gov/fungal/candida-auris/c-auris-antifungal.htm), based on the modal distribution of MICs of approximately in addition to the C. albicans break points.

Lines 53-68, The description of each of the isolates analyzed in lines 53-58, in my opinion, fits better in the results section Origin of Candida auris isolates. Did the patient corresponding to the Cau2 isolate have no history of medical treatment abroad or frequent travel?

Line 150, Table 1, I considered that the authors could omit or encode the name of the hospitals, it would be more interesting to add a column with information about the countries where the patients received medical treatment abroad.

Line 156. The narrative could be improved by presenting the clade identification results first as the resulted clade is mentioned in the description of aggregating phenotype and in the results about antifungal resistance mutations.

Line 172 the Heatmap of the C. auris trials table, shows redundant information with figure 2, I would suggest keeping only figure 2

Line 237 describes mutations in the FKS1 gene, which are not located in the Hotspots regions that have been widely described by association Resistance to echinocandins. This should be brought up in the discussion section.

Line 241 I would move this Phylogeographic clade assignment section to the beginning of the results as section 3.2, just after the description of the isolates

It would be very interesting if the authors decided to complement the phylogenetic analysis by including more clade I isolates from Turkey, India, and Greece and clade III isolates from Spain. Include other isolates with WGS available at SRA NCBI, to assess the relatedness of these Austrian isolates with isolates from the different countries mentioned in the travel anamnesis

Line 300 The results for echinocandin resistance would have a different interpretation if instead of using the breakpoints for Candida albicas, the tentative MIC values for Candida auris in the general guideline proposed by the CDC (https://www.cdc.gov/fungal/ candida-auris/c-auris-antifungal.html) were used. Compared with these tentative breakpoints, all isolates would be susceptible to Echinocandins, which is consistent with the molecular finding of the absence of mutations in the hotspot regions of the FKS1 gene. I believe that the authors should address it in their discussion.

Round 2

Reviewer 1 Report

1) Validate NGS Pipeline

The authors made significant improvements on the bioinformatics analysis. They still have to rename "variants" throughout the manuscript (and also in Table 2) as SNPs.

2) Results section on SNP analysis

Regarding the folllowing sentences: "Isolates Cau1 and Cau2 as well as Cau3 and Cau5 showed a higher degree of relatedness to each other with 44 SNPs and 76 SNPs, respectively. Overall, it is likely that the strains belong to different subclades as they are separated by more than 200 SNPs and more than 900 from the clade I reference strain B8441." It is unclear what the authors mean with "the strains". If is ment "Overall, it is likely that the strains Cau1 and Cau2 belong to different subclades than Cau3 and Cau5, and reference strain B8441, as they are separated by more than 200 and 900 SNPs, respectively." I would agree. Please adapt the sentence to improve clarity. 

For the following sentence "the isolate Cau4 was most closely related to African clade III with 49 variants." adapt to "the isolate Cau4 was most closely related to African clade III with 49 SNPs to clade III reference isolate xxxx." 

3) Discussion: conclusions on SNP data

The authors mention "The isolates Cau1 and Cau2 as well as Cau3 and Cau5 were highly related, although no epidemiological links could be found." Isolates Cau1 and Cau2 as well as Cau3 and Cau5 are indeed more related to each other as compared to other isolates, but the number of SNPs (44 and 76) is still very high. With such high number of SNPs there is certainly no epidemiological link. With less than ~10 SNPs it is worthwhile to look into this. Please adapt or delete sentence.

4) Line 146: something went wrong with a reference.
